# Active Matting

**Xin Yang**[*]
Dalian University of Technology
City University of Hong Kong
xinyang@dlut.edu.cn

**Ke Xu**[*]
Dalian University of Technology
City University of Hong Kong
kkangwing@mail.dlut.edu.cn

**Shaozhe Chen**
Dalian University of Technology
csz@mail.dlut.edu.cn

**Shengfeng He**[†]
South China University of Technology
hesfe@scut.edu.cn

**Baocai Yin**
Dalian University of Technology
ybc@dlut.edu.cn

**Rynson W.H. Lau**[‡]
City University of Hong Kong
rynson.lau@cityu.edu.hk

## Abstract

Image matting is an ill-posed problem. It requires a user input trimap or some strokes to obtain an alpha matte of the foreground object. A fine user input is essential to obtain a good result, which is either time consuming or suitable for experienced users who know where to place the strokes. In this paper, we explore the intrinsic relationship between the user input and the matting algorithm to address the problem of *where* and *when* the user should provide the input. Our aim is to discover the most informative sequence of regions for user input in order to produce a good alpha matte with minimum labeling efforts. To this end, we propose an active matting method with recurrent reinforcement learning. The proposed framework involves human in the loop by sequentially detecting informative regions for trivial human judgement. Comparing to traditional matting algorithms, the proposed framework requires much less efforts, and can produce satisfactory results with just 10 regions. Through extensive experiments, we show that the proposed model reduces user efforts significantly and achieves comparable performance to dense trimaps in a user-friendly manner. We further show that the learned informative knowledge can be generalized across different matting algorithms.

## 1   Introduction

Alpha matting (or image matting) refers to accurately extracting a foreground object of interest from an input image. This problem as well as its inverse process (known as *image composition*) have been well studied by both research and industrial communities. Mathematically, alpha matting can be modeled by the following under-constrained equation:

$$I_z = \alpha_z F_z + (1 - \alpha_z)B_z, \tag{1}$$

where $z = (x, y)$ denotes the pixel position in the input image $I$. $F$ and $B$ refer to the output foreground and background images. $\alpha$ is the alpha matte, whose values range between $[0, 1]$ defining the opacity of the foreground.

---

[*]Joint first authors.
[†]Corresponding author.
[‡]This work was led by Rynson Lau.

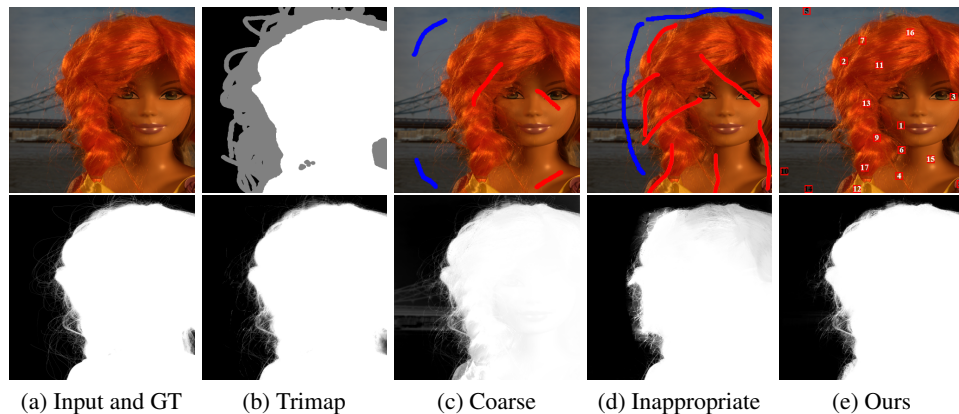

| (a) Input and GT | (b) Trimap | (c) Coarse | (d) Inappropriate | (e) Ours |

Figure 1: Limitations of existing matting approaches. Creating a trimap is tedious, and scribbles may produce unsatisfactory mattes if they are inappropriately placed or not dense enough. The proposed approach is able to generate comparable results to using trimaps, with only a few labelled regions.

This highly ill-posed problem is typically addressed by introducing additional information, in the form of user-defined trimaps [6, 19] or strokes [8, 21], to confine the scale of unknowns. However, the quality of the trimap can largely affect the accuracy of the final output [13, 24], and labeling a fine trimap is also tedious. Although drawing strokes may be a more user-friendly way of obtaining the foreground-background information [11, 13], scribbles provide only a small set of labelled pixels, which can be regarded as a less-accurate trimap and may not be able to provide sufficient constraints to solve Eq. 1 (see Figure 1(c)). More importantly, if scribbles are not correctly drawn, the matting algorithm may be misled to produce unsatisfactory mattes (see Figure 1(d)).

The above observations indicate that a good output matte requires dense labeling (i.e., manual efforts) as well as appropriately placing the labels (i.e., skills). In this paper, we aim to automatically determine "where" to best place the labels to minimize human efforts and the reliance on their proficiency of the matting problem. To this end, we propose to learn to detect the regions that affect the matting performance most. We refer to these regions as *informative regions*. We propose an active matting model with a recurrent neural network to discover these informative regions. The user is involved in the loop, but only required to label a suggested region as foreground or background. This strategy proposes a sequence of regions to the user for labeling one by one, and learns from the user's feedbacks. We adopt the reinforcement learning strategy to allow a direct supervision based on an arbitrary matting algorithm and a ground-truth alpha matte. The proposed network is able to output an accurate matte in a few iterations. Extensive experiments show that our model is a promising solution to address the problem of "where to draw" so as to improve the interaction speed and matting accuracy.

The main contributions of this paper are as follows: 1. We propose an active model to learn to detect informative regions for alpha matting. Our model can actively suggest a small number of informative regions for the user to label. 2. We propose a recurrent network with the reinforcement learning strategy. The network is trained via user feedbacks. 3. We delve into the problem of informative regions for matting, and show that the learned informative knowledge can be generalized to a variety of matting algorithms.

## 2 Methodology

### 2.1 Overview

Our goal here is to explore the problem of *where* to label image regions for obtaining a good matte. Due to the dependency of the subsequent selection of information regions to a user's feedback, we consider it inappropriate to determine all the informative regions at the same time (discussed in Section 3). Instead, we factorize this problem into an interactive search for a sequence of informative

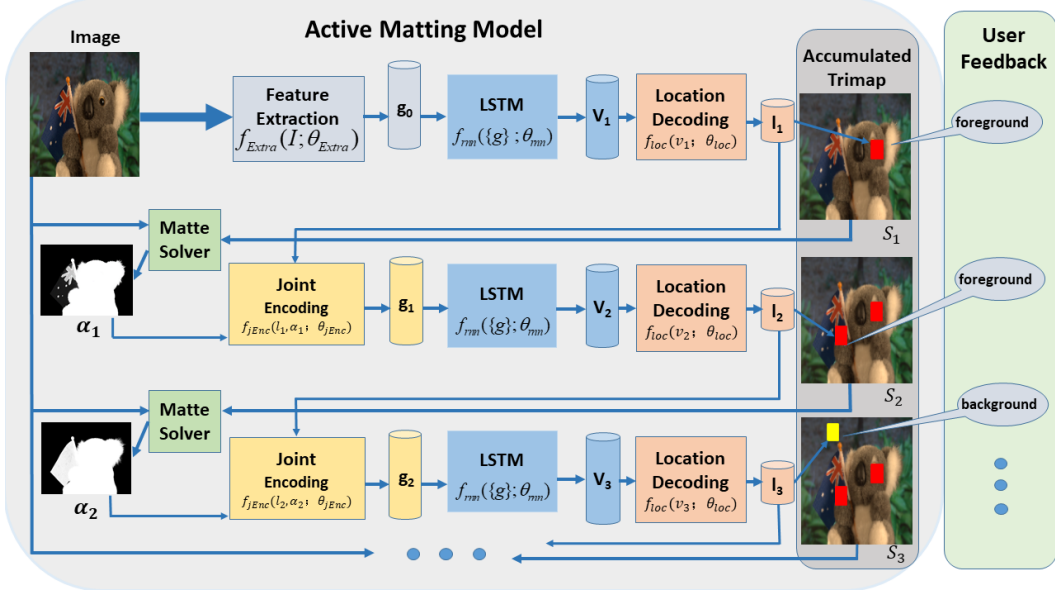

Figure 2: Overview of the proposed active model. (All rectangular boxes of the same color share the same parameters.)

regions. In this way, the image uncertainty problem can be gradually resolved, depending on the image content, previous suggested regions and the corresponding user feedbacks.

We have considered two key problems in our model design: how to detect the next most informative region based on the previously suggested regions and user feedbacks, and how to allow the proposed method to work with arbitrary matting algorithms. We address the first problem by using a **RNN unit**, which is able to make sequential decisions based on previous knowledge, and the second problem by using a matte solver together with our **reinforcement training strategy**.

Figure 2 shows the proposed pipeline, which can be summarized as follows. An input image is first fed into the **Feature Extraction Net** to extract the image features $g_0$. $g_0$ is then fed into the **RNN Unit** to provide the "visual" information for prediction, which is then decoded by the **Location Decoding Net** to obtain the first suggested informative region (represented as a 2D coordinate $l_1$). At each iteration after a region is suggested, our model asks the user to indicate if the region belongs to the foreground or background layer. Each pixel within this region will then be assigned with a corresponding label in the accumulated trimap. The **Matte Solver** takes the input image and the accumulated trimap as input and computes a matte, which is then fed with the 2D coordinate of the previously suggested region to the **Joint Encoding Net** to jointly encode the relationship between the previous region suggestion and the resulting matte. Finally, the **RNN Unit** uses the information that encodes the previous region-matte relationships and the initial visual features to suggest the next informative region for user input. The proposed network learns from the user feedbacks, and we adopt reinforcement learning to assign training reward for each detected informative region.

## 2.2 Architecture

We first present the detailed architecture of our active matting model.

**The Feature Extraction Net.** This network serves as a feature extraction module. It analyzes the input image $I$ and projects it to a lower feature space: $g_0 = f_{Extra}(I; \theta_{Extra})$, where $\theta_{Extra}$ are the network parameters, and $g_0$ is a $1 \times 1000$ vector. In our implementation, we adapt the VGG16 network [18] here without its final softmax layer.

**The RNN Unit.** We use the Long Short Term Memory (LSTM) unit [10] to fuse the image features and the region-matte relations to predict the location for the next informative region: $v_{i+1} = f_{rnn}(\{g_k\}; \theta_{rnn})$, where $k = 1, 2, ...i$, and $\theta_{rnn}$ represents the parameters. $v_i$ is a 128-dimension

vector. In this way, our model can suggest the next region by considering all the previously suggested regions and their resulting mattes.

**The Location Decoding Net.** It takes the predicted information $v_i$ from the RNN Unit and decodes it into a 2D coordinate: $l_i = f_{loc}(v_i; \theta_{loc})$, where $i$ is the iteration number, and $\theta_{loc}$ refers to the network parameters. Our model actively proposes $20 \times 20$ regions centered at $l_i$.

**The Matte Solver.** The accumulated trimap $s_i$ is generated by the current and all previous pairs of suggested regions and their corresponding user inputs. We pass $s_i$ and image $I$ through the matte solver to obtain the latest matte $\alpha_i$: $\alpha_i = f_{solver}(s_i, I)$. In our implementation, we use shared matting [7] as the matte solver because it is efficient for training. However, as will be shown in Section 3, the sequence of informative regions produced by our model is general and can be used with any matting algorithms.

**The Joint Encoding Net.** Once we have a proposed 2D coordinate $l_i$, the joint encoding net fuses this coordinate with its corresponding matte, for the purpose of establishing the relation between the suggested region and the resulting matte. The relation is encoded as $g_i = f_{jEnc}(l_i, \alpha_i; \theta_{jEnc})$, where $\theta_{jEnc}$ refers to the network parameters. Similar to $g_0$, $g_i$ is also a $1 \times 1000$ vector. In our implementation, the joint encoding net takes the estimated coordinates and a local patch of the resulting matte centered at the estimated coordinates as input, and outputs a vector for the following RNN Unit. The patch size is fixed at $75 \times 75$.

### 2.3 Reinforcement Learning of the Sequence

The proposed active model needs to establish a connection between the matting solver, the suggested regions, and the ground truth matte. To obtain supervision for the proposed model, each suggested region (*i.e.*, the accumulated trimap $s_i$) is fed to the matting solver, and its output matte is then compared with the ground truth matte. However, this training process involves a matting solver, which is non-differentiable. Thus, we are not able to train it using traditional back-propagation strategies, and we take an alternative solution with reinforcement learning [22].

We first measure the accuracy of the alpha matte using the Root Mean Square Error (RMSE) metric:

$$RMSE = \sqrt{\frac{1}{N} \sum_{z=1}^{N} \left\| \hat{\alpha_z} - \alpha_z^{gt} \right\|_2^2}, \tag{2}$$

where $N$ refers to the total number of pixels. $\hat{\alpha}$ and $\alpha^{gt}$ represent the estimated and ground truth mattes, respectively. Assuming that the alpha values have a Gaussian distribution, minimizing Eq. 2 is approximately equivalent to maximizing $\log p(\alpha|I, \Theta)$, where $\Theta$ refers to the model parameters, and $p(\cdot)$ refers to the likelihood function. We note that this log-likelihood term $\log p(\alpha|I, \Theta)$ can be maximized by marginalizing over a sequence of proposed regions $\{l_i\}$: $\log p(\alpha|I, \Theta) = \log \sum_l p(l|I, \Theta) p(\alpha|l, I, \Theta)$. This marginalized function can be learned by maximizing its lower bound as discussed in [3]:

$$F = \sum_l p(l|I, \Theta) \log p(\alpha|l, I, \Theta). \tag{3}$$

By taking the derivative of Eq. 3 w.r.t. the model parameters $\Theta$, we obtain the training rules as:

$$\frac{\partial F}{\partial \Theta} = \sum_l p(l|I, \Theta) \left[ \frac{\partial \log p(\alpha|l, I, \Theta)}{\partial \Theta} + \log p(\alpha|l, I, \Theta) \frac{\partial \log p(l|I, \Theta)}{\partial \Theta} \right]. \tag{4}$$

Note that the first derivation in Eq. 4 is the gradient of the matte estimation with respect to the model parameters. We have designed our matte solver as a plug-and-play module, and excluded this term from our final loss function. In this way, our model can focus on the task of finding the informative regions, and will not be influenced by the matting method used, allowing the suggested regions to be sufficiently general for different matting methods.

To avoid an exponentially growing solution space of location $l$, we then adopt Monte Carlo sampling for approximation: $\tilde{l}_i^m \sim p(l_i|I, \Theta) = N(l_i; \hat{l}_i, \Omega)$, where $\hat{l}_i$ is the estimated location at the $i$-th iteration and $\Omega$ is a predefined standard deviation. Hence, Eq. 4 can be rewritten as:

$$\frac{\partial F}{\partial \Theta} = \frac{1}{M} \sum_{m=1}^{M} \sum_{i=1}^{T} \left[ \log p(\alpha|\tilde{l}_i^m, I, \Theta) \frac{\partial \log p(\tilde{l}_i^m|I, \Theta)}{\partial \Theta} \right], \tag{5}$$

where $M$ is the training episode, and $T$ is the total number of proposals at each episode.

The last problem that we need to solve is that the log-likelihood $\log p(\alpha | \tilde{l}_i^m, I, \Theta)$ may introduce unbounded high variance to the gradient estimator on bad regions suggested during training. This can be addressed by introducing the gradient variance reduction strategy [12] to our model. We replace the log-likelihood with a difference value between the reward function $R$ and the output of a baseline network: $b_i = f_{base}(v_i; \theta_{base})$, where $v_i$ is the output of the RNN unit at the $i$-th iteration and $\theta_{base}$ refers to the network parameters. The baseline network is trained to learn the expected value of $R$ in order to normalize the reward to be mean zero [20]. Hence, the training loss is formulated as:

$$\frac{\partial F}{\partial \Theta} \approx \frac{1}{M} \sum_{m=1}^{M} \sum_{i=1}^{T} (R_i^m - b_i) \frac{\partial \log p(\tilde{l}_i^m | I, \Theta)}{\partial \Theta}. \tag{6}$$

In this way, we factorize the goal of maximizing the reduction in RMSE in a stepwise manner, which means that our network is able to propose a sequence of regions and each will lead to a maximum decline in RMSE. In other words, these suggested, ordered regions are informative regions for the matting problem.

**Training Strategy and Reward Definition.** At the $i$-th iteration, our model suggests a region $l_i$ and receives a reward $R_i$, which indicates the quality of $l_i$. Suppose that there are $S$ possible locations to be selected for suggestion at the $i$-th iteration, we can dynamically define the reward by how much the $j$-th region affects the estimated matte as:

$$R_i^j = \frac{\|\alpha_{i-1}^j - \alpha^{gt}\| - \|\alpha_i^j - \alpha^{gt}\|}{\max\{\|\alpha_{i-1} - \alpha^{gt}\| - \|\alpha_i^{\{S\}} - \alpha^{gt}\|\}}. \tag{7}$$

where the region corresponding to the maximum RMSE reduction is regarded as the ground truth region suggestion and will receive a reward of $1$. The other regions will receive rewards according to their percentages of RMSE reduction.

To train our model at the $i$-th iteration, we run our network $S$ times repeatedly and then collect $S$ latent regions as suggestions for the $i$-th iteration. Each latent region receives a reward according to the RMSE reduction. Based on the given rewards, we select the best one as the suggested region to facilitate the training of finding informative regions. S cannot be set to 1. Otherwise, the proposed model fails to distinguish what a good region is, as it always receives a reward of 1 no matter how much RMSE declines. We empirically compare the performance by selecting $S$ from $\{1, 2, 5\}$. While $S = 2$ performs $\sim 30\%$ better than $S = 1$, $S = 5$ performs only $\sim 1.3\%$ better than $S = 2$ but the computation time is double. In addition, a large value (*e.g.*, $S = 10$) will lead a very costly training. As a result, we set $S$ to 2 during training, which achieves a good balance between accuracy and efficiency. We fix $S$ to 1 during inference.

During training, each suggested region is answered automatically according to the ground truth matte. Specifically, we only provide two possible answers, *i.e.*, foreground or background. If a suggested region contains unknown pixels, we simply skip that iteration and our model will not receive a reward. In this way, we want our model to focus on the foreground/background informative regions only, which are easy for the user to label. In the test stage, our model behaves as a fully forward model, which actively suggests regions for user feedbacks.

## 3 Experiments

### 3.1 Experimental Setup

**Experiment Environment.** Our active matting model is implemented using Tensorflow [1], and trained and tested on a PC with an i7-6700K CPU, 8G RAM and an NVIDIA GTX 1080 GPU. The input images are resized to $400 \times 400$. Generally, it takes about $0.35s$ for our model to suggest a region after receiving the user feedback. We train our network from scratch with the truncated normal initializer. The learning rate is set to $10^{-3}$ initially and then goes through an exponential decay to the minimum learning rate $10^{-4}$. We also crop the gradients to prevent gradient explosion.

**Experiment Datasets.** We have conducted our experiments on three challenging datasets, the portrait dataset [17], the matting benchmark [14], and our rendered-100 dataset. The portrait dataset

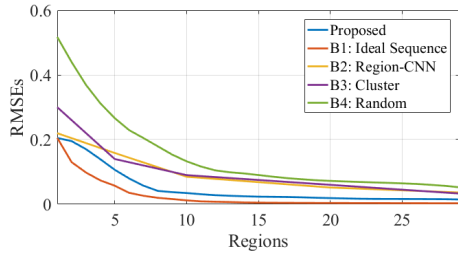

(a) Evaluation on the matting benchmark [14]

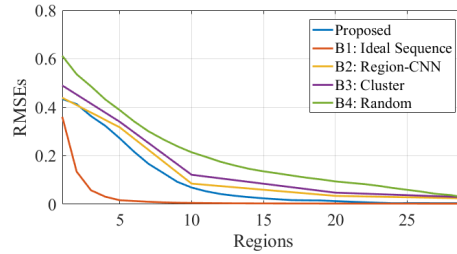

(b) Evaluation on the portrait dataset [17]

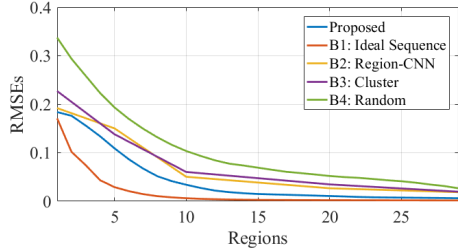

(c) Evaluation on the Rendered-100 dataset

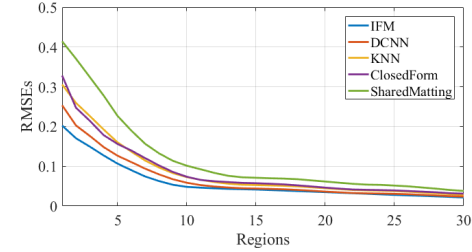

(d) Ours method applied in different matting algorithms

Figure 3: Evaluation of the proposed method. (a)-(c) Comparison with the four baselines on three datasets. (d) The learned informative knowledge is used with different matting algorithms. IFM refers to [2], DCNN refers to [5], KNN refers to [4], ClosedForm refers to [11], and SharedMatting refers to [7].

contains 1,700 training images, 300 testing images, and their corresponding ground truth mattes. The matting benchmark consists of 27 images with user-defined trimaps and ground truth mattes, and 8 images without trimaps nor mattes. We use the portrait testing images and 27 images of the matting benchmark for evaluation.

We train our model using the training set of the portrait dataset. To avoid overfitting, we propose a rendered-100 dataset for fine tuning, which has 100 images and their corresponding ground truth mattes. We use 90 images for fine tuning with data augmentation, and 10 images for testing. To build the rendered-100 dataset, we select different 3D models as foreground objects (e.g., bunny, hairball and metal sphere), and use natural images as backgrounds. In particular, we select foreground objects with thin structures (*e.g.*, furs and leaves), or having similar textures with the background, to simulate challenging scenarios. We show sample images in our supplemental. The complete render-100 dataset (including rendered images, extracted foreground objects, and ground-truth alpha mattes) can be found at [4].

**Four Baselines.** We have constructed the following baselines in our experiments:

**B1:** Since we sequentially generate the informative regions, and their effectiveness is measured by the RMSE between the produced alpha matte and the ground truth matte, there must be an *ideal sequence* that would produce the minimum RMSE (or maximum RMSE reduction) at each step. We exhaustively search every region in the image at each step. It takes a few hours per image in order to obtain this ideal sequence, which can be used to show the upper bound performance.

**B2:** This baseline is to compute the informative regions without the recurrent mechanism. We train seven CNNs such that each of them computes 1, 5, 10, 15, 20, 25 or 30 regions simultaneously. We refer to them as Region-CNNs. Each of them takes an image as input, encodes the extracted features to a $2 \times N$ vector using a fully-connected layer, and then outputs $N$ coordinates representing $N$ regions. They are all trained on the top-$N$ ideal regions, where $N$=1, 5, 10, 15, 20, 25 or 30.

Table 1: RMSE comparison of different input strategies with different matting methods. ("**Active**" refers to our sequence of 20 informative regions, while "Ideal Sequence" refers to the sequence of 20 ground truth regions.) We report the RMSEs of 8 example images and the average RMSEs for all 27 images from the matting benchmark [14]. The best results of different types of input are shown in gray background. The time costs (seconds) are shown below Trimap+IFM [2] and **Active**+IFM [2].

| | 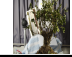 |  |  |  |  |  |  | Mean RMSEs |
|---|---|---|---|---|---|---|---|---|
| Trimap+IFM [2] | 0.094 181s | 0.046 209s | 0.021 195s | 0.018 192s | 0.051 215s | 0.017 187s | 0.022 205s | 0.031 204s |
| Ideal Sequence+IFM [2] | 0.096 | 0.048 | 0.023 | 0.019 | 0.064 | 0.019 | 0.022 | 0.032 |
| Trimap+DCNN [5] | 0.117 | 0.049 | 0.021 | 0.019 | 0.079 | 0.019 | 0.021 | 0.033 |
| Ideal Sequence+DCNN [5] | 0.121 | 0.051 | 0.022 | 0.022 | 0.081 | 0.021 | 0.021 | 0.034 |
| **Active**+IFM [2] | 0.099 55s | 0.051 66s | 0.024 58s | 0.019 62s | 0.071 54s | 0.021 55s | 0.024 53s | 0.034 57s |
| **Active**+DCNN [5] | 0.121 | 0.052 | 0.024 | 0.023 | 0.081 | 0.021 | 0.023 | 0.035 |
| Trimap+Shared [7] | 0.104 | 0.059 | 0.023 | 0.021 | 0.071 | 0.024 | 0.022 | 0.035 |
| **Active**+Shared [7] | 0.111 | 0.062 | 0.027 | 0.023 | 0.078 | 0.026 | 0.024 | 0.039 |
| Trimap+KNN [4] | 0.097 | 0.053 | 0.029 | 0.031 | 0.066 | 0.022 | 0.037 | 0.043 |
| Trimap+Global [9] | 0.111 | 0.059 | 0.024 | 0.023 | 0.072 | 0.018 | 0.024 | 0.045 |
| **Active**+KNN [4] | 0.102 | 0.058 | 0.034 | 0.035 | 0.074 | 0.023 | 0.044 | 0.049 |
| Trimap+Learning [23] | 0.126 | 0.107 | 0.028 | 0.015 | 0.084 | 0.022 | 0.015 | 0.051 |
| Trimap+ClosedForm [11] | 0.136 | 0.117 | 0.029 | 0.016 | 0.097 | 0.023 | 0.016 | 0.056 |

**B3:** We develop this baseline based on the clustering strategy. The input image is divided into 5, 10, 15, 20, 25 and 30 clusters based on the pre-defined features suggested in [16]. After that, the center grids of each cluster are proposed for user labeling.

**B4:** We use the randomly generated sequence as another baseline.

### 3.2 Comparison with Baselines

Figure 3(a-c) compare the proposed method with four baselines in terms of root mean squared error (RMSE). Results show that the proposed method has comparable performance to the ideal baseline **B1** on three datasets, particularly after 15 iterations. Although the proposed method is trained on the extremely unbalanced datasets (1,700 portrait and 90 rendered images), it still shows good generalization to natural images on the matting benchmark.

We also validate the proposed method against two baselines that directly generate multiple informative regions. The first uses CNNs to automatically learn the features for predicting regions (**B2**) and the second uses hand-crafted features with a clustering strategy (**B3**). Figure 3(a-c) show that our model with sequential learning performs better than these two baselines. This is because generating multiple informative regions simultaneously does not consider the dependency between informative regions and user feedbacks. While some regions may be informative by themselves, they may provide similar information to the matting solver. Sequentially generating regions produce complementary results. Finally, we can see that the learned informative knowledge outperforms random sequence (**B4**).

### 3.3 Comparison with Trimaps

Here, we show the comparison between using a fine trimap and 20 informative regions as input. Different matting algorithms are fed with a trimap or our informative regions. The ideal sequence is also shown for reference.

In order to have a fair comparison on generating the trimap in terms of interaction time, we asked 10 users to generate a trimap for each image from scratch. Specifically, users were first asked to draw scribbles to indicate foreground/background and a bounding box of the foreground object. Grabcut [15] was then used to generate a trimap based on the user input. Regions in the trimap with low confidence were set to unknown regions. Users were able to iteratively refine the input trimap based on the alpha matting result. This process stopped when the users were satisfied with the output

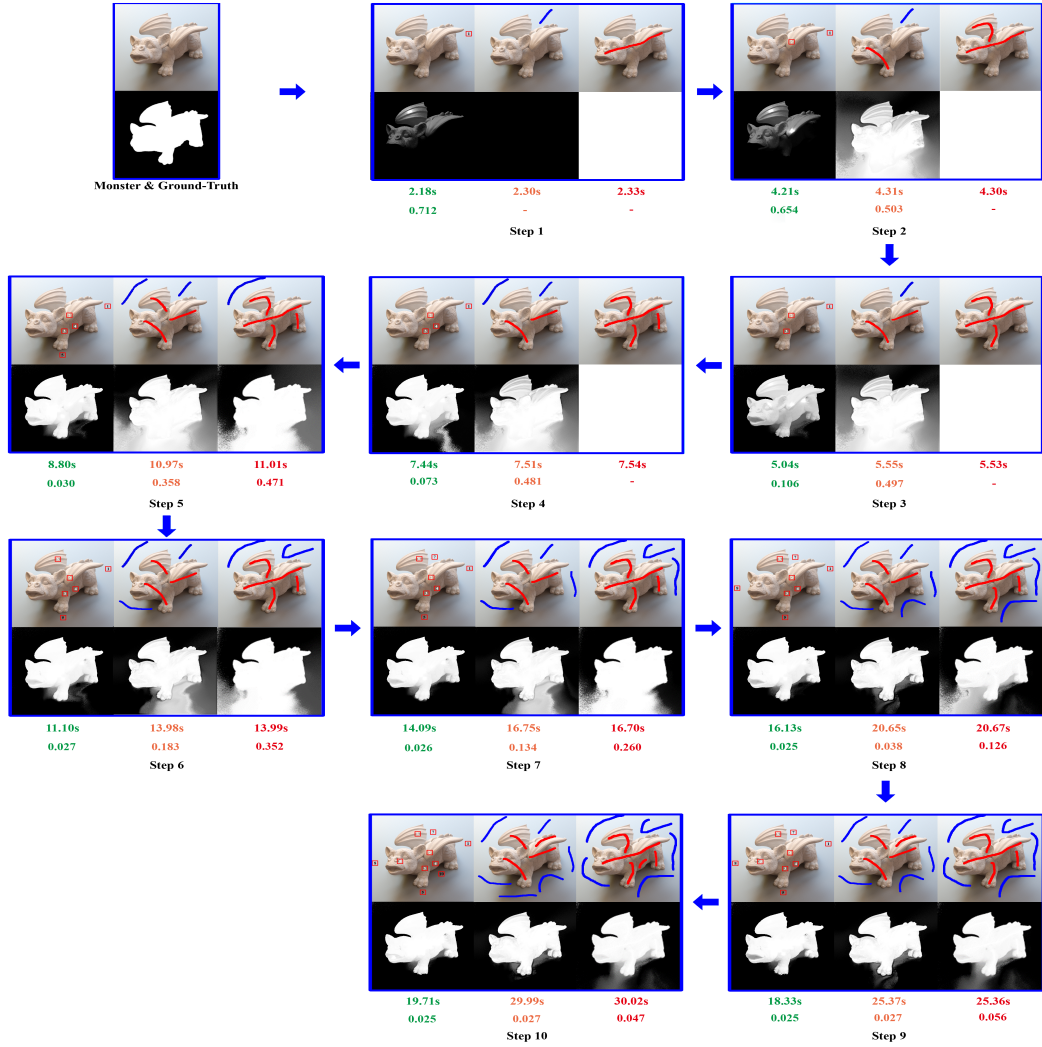

Figure 4: Step-wise comparison between our active model and user scribbles. In each step, the left column is generated by our active matting, the middle and right columns are the scribbles drawn by the experienced and inexperienced users, respectively. We report the labeling time (in seconds) and RMSE for each iteration.

matte. The time taken for the entire process was recorded, including drawing, Grabcut computation, matting computation and refinement. The average RMSEs and times are reported.

Table 1 shows the comparison of different input strategies (i.e., trimaps, ideal sequence, and the proposed sequence). We report 8 images from the matting benchmark [14] as well as the average performance on the whole matting benchmark (27 images). The interaction time cost is reported below the best methods of the trimap input and active matting. Note that the proposed method has not been trained on these images. Results show that the proposed 20 informative regions yield comparable matting performance to a fine-labelled trimap in terms of quality. On the other hand, generating a fine trimap costs much users efforts, and it takes about 3 minutes to obtain a good alpha matte. Instead, our active model is free from trimaps, and it takes around 1 minute from feeding an image input to getting an alpha matte. We show that the proposed informative regions can achieve comparable results to the trimap-based method with less labeling efforts.

Furthermore, although we use shared matting [7] for computing mattes during training, results show that our informative regions are general across different matting algorithms. As shown in Figure 3(d), the RMSEs drop significantly with the first 5 informative regions for all the methods,

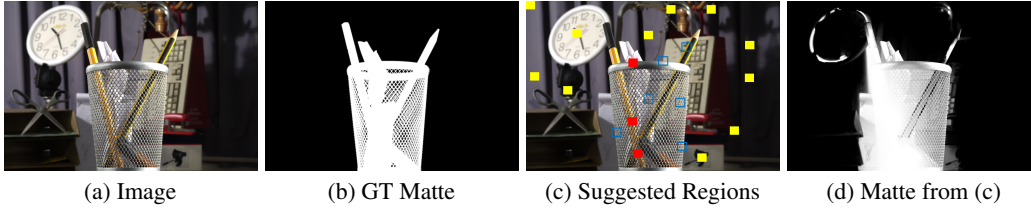

| (a) Image | (b) GT Matte | (c) Suggested Regions | (d) Matte from (c) |

Figure 5: A failure case. The input image in (a) contains thin structures that are smaller than the size of the suggested regions (red boxes in (c)), resulting in an unsatisfactory matte as shown in (d).

and the subsequent regions gradually refine the resulted alpha mattes. This result implies that our informative regions are independent of the matting algorithms used and can be used as an image feature for other applications.

### 3.4 Comparison with Scribbles

We conduct a stepwise comparison with user scribbles. As shown in Figure 4, the active model proposes 10 regions to a user (one with little knowledge about matting) for labeling, and two users (one is experienced user familiar with matting and the other is inexperienced) are asked to draw 10 scribbles on the image for comparison. The results of each step and the interaction time are shown. At the beginning (step 1), the proposed active model can accurately find the right region such that the matting algorithm [7] can distinguish the foreground and background layers by color distributions. On the contrary, both the scribbles from the users are not sufficient to generate the mattes. Moreover, the proposed model can generate satisfactory result within 5 steps, while the results from scribbles still suffer from background-foreground separation. Our resulting matte is further refined in the subsequent 5 steps and achieves better alpha matte with less labeling time than using scribbles.

## 4 Conclusion and Limitation

In this paper, we propose a novel active model for matting. Our model can actively find and propose informative regions for users to label the foreground or background layer. After several question-and-answer iterations, a high quality matte can be obtained with minimum human efforts. We integrate the idea of informative regions to the matting process by formulating it as a reinforcement learning problem. The proposed informative regions are general across different matting algorithms.

The limitation of our active matting model is that it does not perform well on thin structures. As shown in Figure 5, the foreground object contains thin structures that are smaller than the suggested regions, leading to an unsatisfactory result. A possible solution to this problem is to incorporate scribbles, in addition to suggested regions, for labeling fine details. This will be left as a future work.

For further research, we plan to extend our model to a realtime scribbling-based system, *i.e.*, providing suggestions to the user directly on the image with the probability of being informative in realtime. This provides meaningful guidance for users to allow them to directly draw scribbles on the most informative regions. On the other hand, we plan to develop a differentiable model using convolutional neural networks, instead of existing matting solvers, to fully exploit the information obtained from the proposed informative regions.

## 5 Acknowledgements

We thank the anonymous reviewers for the insightful and constructive comments, and NVIDIA for generous donation of GPU cards for our experiments. This work is in part supported by an SRG grant from City University of Hong Kong (Ref. 7004889), NSFC grants from National Natural Science Foundation of China (Ref. 91748104, 61632006, 61425002, 61702194), and the Guangzhou Key Industrial Technology Research fund (No. 201802010036).

## Footnotes

[4] http://www.cs.cityu.edu.hk/~rynson/projects/matting/ActiveMatting.html

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
