[Supplementary Material]

# Active Matting (Supplementary Material)

**Xin Yang**[*]
Dalian University of Technology
City University of Hong Kong
xinyang@dlut.edu.cn

**Ke Xu**[*]
Dalian University of Technology
City University of Hong Kong
kkangwing@mail.dlut.edu.cn

**Shaozhe Chen**
Dalian University of Technology
csz@mail.dlut.edu.cn

**Shengfeng He**[†]
South China University of Technology
hesfe@scut.edu.cn

**Baocai Yin**
Dalian University of Technology
ybc@dlut.edu.cn

**Rynson W.H. Lau**[‡]
City University of Hong Kong
rynson.lau@cityu.edu.hk

## 1 Introduction

This supplementary material provides: (a) the analysis of the hyper-parameters used in the proposed active matting model, including the size of proposed regions and the number of regions for users to label (Section 2), (b) more qualitative results of the proposed active matting (Section 3), and (c) sample images of our rendered-100 dataset (Section 4).

## 2 Analysis of the Hyper-parameters

The size of the proposed regions is the most important hyper-parameter, as it is sensitive to two kinds of image scenes. The first scenario contains small transparent structures, e.g., furs and hair (Figures 1(a) and 1(b)). The other scenario contains isolated unknown layers within the foreground or background layers (e.g., Figures 1(c) and 1(d)).

|     |     |     |     |
| --- | --- | --- | --- |
| (a) | (b) | (c) | (d) |

Figure 1: Sample images of two typical scenarios from the benchmark [2] and portrait dataset [3]. (a) and (b) contain small transparent structures, while (c) and (d) contain isolated regions.

---

[*]Joint first authors.

[†]Corresponding author.

[‡]This work was led by Rynson Lau.

Based on these two different types of scenes, we conduct the experiments with different settings. The size of the proposed regions is set to $40 \times 40$, $20 \times 20$, and $16 \times 16$. We also compare 10, 20 and 30 regions to find out how many regions are enough for computing satisfactory mattes.

The results are shown in Figure 2. We observe that larger region size performs better in the images with small transparent structures, while smaller region size performs better in the images with isolated unknown regions. This is mainly because the isolated unknown regions contain lots of small regions that cannot be covered by the proposed region. In addition, generating more regions yields better results but will cost more labeling efforts. To get a good balance between different scenes, labeling efforts, and the matting quality, our model is set to propose 20 regions with a size of $20 \times 20$.

(a) Small transparent structures

(b) Isolated unknown regions

Figure 2: Histograms comparisons of different scenes (*i.e.*, small transparent structures and isolated unknown regions). Groups 1-3 represent the sizes of $40 \times 40$, $20 \times 20$, and $16 \times 16$, respectively.

# 3   More Experimental Results

In this section, we present more qualitative evaluation of our active matting. We first show the effectiveness of the top 5 informative regions in Figure 3, referred to as **Fast Active Matting**. We also show the results of 20 informative regions proposed by the active matting model, referred to as **High-Quality Active Matting**. Generating more regions help refine the mattes. Finally, we show that our active matting model can generate a competent alpha matte combined with a single scribble, referred to as **Incorporating Human Prior with a Single Scribble**.

## 3.1   Fast Active Matting

Figure 3 shows some matting results generated by the top-5 regions suggested by the proposed model. It is useful particularly when the user wants a quick result rather than an accurate one.

## 3.2   High-Quality Matting

Although top-5 regions can achieve a reasonable alpha matte, the user can continue to label more regions to obtain a fine matte. Figure 4 shows sample images and the resulting mattes with 20 regions. We can see that mattes with 20 regions are visually very close to the ground truth mattes.

## 3.3   Incorporating Human Prior with a Single Scribble

In some challenging scenes (*e.g.*, overlapped foreground/background), actively searching the informative regions after 20 iterations may not help refine the local fine details significantly. Therefore, it is beneficial to incorporate human prior in the form of a single scribble in the early stages. Figure 5 shows one sample, where we can see that the alpha matte can be refined with this simple interaction. More sophisticated way to incorporate human prior will be studied in the future work.

# 4   Rendered-100 Dataset

As the ground-truth alpha mattes is difficult to obtain, we create a dataset containing pairs of images and alpha mattes using the Physically Based Rendering Technology (PBRT) [1]. Figure 6 shows

Figure 3: Four groups of **Fast Active Matting** sample images. Each group consists of the input image (top left), regions proposed by active matting (top right), ground truth matte (bottom left) and the resulting matte (bottom right).

several samples from our rendered-100 dataset. The full Rendered-100 dataset can be found at `http://www.cs.cityu.edu.hk/~rynson/projects/matting/ActiveMatting.html`.

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

Figure 4: Four groups of **High-Quality Active Matting** sample images. Each group consists of the input image (top left), regions proposed by active matting (top right), ground truth matte (bottom left) and the resulting matte (bottom right).

Figure 5: Illustration of incorporating a single scribble into the proposed method to balance the user efforts and matte accuracy. (a) and (b) show the top-5 regions proposed by our method and the resulting matte. (c) and (d) show the top-5 regions plus one scribble and the corresponding matte.

Figure 6: Sample images from the proposed rendered-100 dataset.