[Reviews · NeurIPS 2018]

Reviewer 1



The authors propose a method for learning how to propose informative regions to be labelled for matting. The model is based on an RNN that uses a matting solver as a black box. Given that in principle the matting solver is non-differentiable, the authors rely on posing the training process as a reinforcement learning problem. The problem is interesting, and the overall idea of making the model learn to propose informative regions is nice. However the execution of the idea is unsatisfactory for the following reasons: The reinforcement learning approach described is based on the direct application of the algorithm developed in ref [3] in the paper. However, this approach is not directly applicable here. The way the adaptation of this approach is done for the matting application is not clear: - In particular in line 121 it is said that the first term of eq. 4 is dropped. That is equivalent to state that p(\alpha | ...) is not at all dependent on \Theeta. That is a key part of the derivation of the algorithm that seems to not hold with other parts of the paper. - Related to the previous point, how is the accumulated trimap, s_i, taken into account in the reinforcement learning process? It does not appear in Sec. 2.3 at all. - Line 114 states that minimizing eq. 2 is equivalent to maximizing log p(...), but this is only right if assuming that p is Gaussian. Regarding the experimental section, for B2 it is not explained how each CNN produces n different regions at once. Some parts of the paper lack clarity or contain confusing information. Some examples: - According to line 91, it seems that l_i is a pixel, but later (e.g. line 116) it is called region, which is rather misleading. - When explaining the architecture in Sec. 2.2, it would help the reader know which space the feature representations are contained in (e.g. is g_0 a 2D, or 3D tensor?). - In line 120, when it is stated that "Note that the first derivation in Eq. 4 is", the authors may mean the first term of the r.h.s. of eq. 4. In summary, it is an interesting problem, but the paper contains too many undefined points as to be considered for NIPS.

Reviewer 2



Due to the importance of the user input trimap or strokes in image matting, this paper propose an active matting method with recurrent reinforcement learning. The proposed model can actively suggest a small number of informative sequence of regions for user to label, which minimizes the labeling efforts. Experimental results show that the proposed model achieves comparable performance to dense trimaps. In addition, the learned informative knowledge can be applied to a variety of matting algorithms as shown in this paper. Strengths: 1. This paper proposes a novel an helpful concept named informative regions to denote the regions that affect the matting performance most. 2. The proposed model is in a user-friendly manner that users are only required to label a suggested region as foreground or background. 3. Extensive results show that the learned informative regions achieve comparable matting performance to a fine-labelled trimap in terms of quality while the proposed method is much faster. 4. The proposed model have well applicability to a variety of matting algorithms. 5. The details are well clarified and the method can be reproduced according to this paper. Weaknesses: 1. The authors run their network S times repeatedly and collect S latent regions as suggestions each iteration. They set S to 2 to achieve a good balance between accuracy and efficiency. More details are missing to support this choice. 2. Several important references are missing. For example, 1) Ning Xu, Brian Price, Scott Cohen, Thomas Huang, Deep Image Matting. CVPR 2017. the authors are encouraged to add comparisons. 2) Liang Yang, Xiaochun Cao, Di Jin, Xiao Wang, Dan Meng: A Unified Semi-Supervised Community Detection Framework Using Latent Space Graph Regularization. IEEE Trans. Cybernetics 45(11): 2585-2598 (2015) where a similar "active" must-link priors are utilized. 3) Chuan Wang, Hua Zhang, Liang Yang, Xiaochun Cao, Hongkai Xiong: Multiple Semantic Matching on Augmented N-Partite Graph for Object Co-Segmentation. IEEE Trans. Image Processing 26(12): 5825-5839 (2017) where co-matting is addressed. 3. In my point of view, the most important significance of this paper is that is can help establishing large-scale matting dataset, which is rarely mentioned in this paper.

Reviewer 3



This paper studies the problem of matting. Given an image, a matting algorithm outputs a per-pixel mask, describing whether each pixel belongs to foreground (1) or background (0). Solving matting requires a user to *actively* interact with the segmentation system, providing supervision in the form of trimap or scribbles. Such interaction is often laborious and requires expertise (i.e. knowing where to draw is best for the specific segmentation algorithm). This paper simplifies the interaction by freeing users from making decisions about where to draw and the drawing action. Instead, users only have to answer a binary question each time, regarding whether the region queried by the algorithm belongs to the foreground or not. To come up with the next region to query at each step, the algorithm takes into account regions queried previously, the current matte, and image content. To learn an end-to-end system that takes an image as input and allow a user to interact with a black-box matte solver, this paper seeks out reinforcement learning. Results show that with the same number of steps of interaction, active matting produces alpha matte with better quality while taking less time than an experienced human user. Here are my questions after reading the paper: - No negative results are presented in the paper. I am curious what the failure cases are for the proposed method. - Is a binary answer ever insufficient for a 75x75 region? Does the algorithm ever propose regions that land on a thin-structure (e.g. hair)? - If I understand the first baseline (i.e. B1 at line 182) correctly, the ideal sequence describes a locally optimal solution, instead of a globally optimal solution. In some sense, that is not an upper bound for the proposed method. Is that right?